

# Continuous measurements of nitrous oxide isotopomers during incubation experiments

Malte Winther[1], David Balslev-Harder[1,2], Søren Christensen[3], Anders Priemé[4,5], Bo Elberling[5], Eric Crosson[6], and Thomas Blunier[1]

[1]Centre for Ice and Climate, Niels Bohr Institute, University of Copenhagen, Denmark
[2]DFM - Danish National Metrology Institute, Kgs. Lyngby, Denmark
[3]Section for Terrestrial Ecology, Department of Biology, University of Copenhagen, Denmark
[4]Section for Microbiology, Department of Biology, University of Copenhagen, Denmark
[5]Center for Permafrost, Department of Geosciences and Natural Resource Management, University of Copenhagen, Denmark
[6]Picarro Inc, Santa Clara, CA 95054 USA

*Correspondence to:* Malte Winther (malte.winther@nbi.ku.dk)

**Abstract.** Nitrous oxide ($N_2O$) is an important and strong greenhouse gas in the atmosphere. It is produced by microbes during nitrification and denitrification in terrestrial and aquatic ecosystems. The main sinks for $N_2O$ are turnover by denitrification and photolysis and photo-oxidation in the stratosphere. In the linear N=N=O molecule $^{15}N$ substitution is possible in two distinct positions, central and terminal. The respective molecules, $^{14}N^{15}N^{16}O$ and $^{15}N^{14}N^{16}O$, are called isotopomers. It has

been demonstrated that $N_2O$ produced by nitrifying or denitrifying microbes exhibits a different relative abundance of the isotopomers. Therefore, measurements of the site preference (difference in the abundance of the two isotopomers) in $N_2O$ can be used to determine the source of $N_2O$ i.e. nitrification or denitrification. Recent instrument development allows for continuous position dependent $\delta^{15}N$ measurements at $N_2O$ concentrations relevant for studies of atmospheric chemistry. We present results from continuous incubation experiments with denitrifying bacteria, *Pseudomonas fluorescens* (producing and

reducing $N_2O$) and *Pseudomonas chlororaphis* (only producing $N_2O$). The continuous analysis of $N_2O$ isotopomers reveal the transient pattern ($KNO_3$ to $N_2O$ and $N_2$, respectively). We find bulk isotopic fractionation of -5.01 ‰ $\pm$ 1.20 for *P. chlororaphis*, in line with previous results for production from denitrification. For *P. fluorescens*, the bulk isotopic fractionation during production of $N_2O$ is -52.21 ‰ $\pm$ 9.28 and 8.77 ‰ $\pm$ 4.49 during $N_2O$ reduction.

   The SP isotopic fractionation for *P. chlororaphis* is -3.42 ‰ $\pm$ 1.69. For *P. fluorescens*, the calculations result in SP isotopic

fractionation values of 5.73 ‰ $\pm$ 5.26 during production of $N_2O$ and 2.41 ‰ $\pm$ 3.04 during reduction of $N_2O$. We interpret the slightly increased isotopic fractionation during reduction to diffusive isotopic fractionation and a difference in active enzymes during production of $N_2O$. In summary, we implemented continuous measurements of $N_2O$ isotopomers during incubation of denitrifying bacteria and believe that similar experiments will lead to a better understanding of denitrifying bacteria and $N_2O$ turnover in soils and sediments and ultimately hands-on knowledge on the biotic mechanisms behind greenhouse gas exchange

of the globe.

**Keywords**



Nitrous oxide, isotopomers, site preference, greenhouse gas, denitrification, *Pseudomonas fluorescens*, *Pseudomonas chlororaphis*

# 1 Introduction

The atmospheric concentration of nitrous oxide ($N_2O$) has increased from approximately 271 ppb before the industrialization to 324 ppb in 2011 (Ciais et al., 2013). This increase has resulted in (1) $N_2O$ being the third most important greenhouse gas, that is $N_2O$ has the third highest contribution to the radiative forcing of the naturally occurring greenhouse gases (Hartmann et al., 2013), and (2) an increased production of nitrogen oxides (NOx) in the stratosphere and thereby an increased ozone-depletion (Forster et al., 2007; Kim and Craig, 1993).

Ice core records show that $N_2O$ concentrations positively correlate with northern hemispheric temperature variations, e.g. during the last glacial-interglacial termination as well as over the rapid climate variations occurring during the glacial period, known as Dansgaard-Oeschger events (D-O events) (Schilt et al., 2010). However, occasionally (e.g. D-O event 15 and 17) the $N_2O$ concentration increases long before the onset of the dramatic temperature change (Schilt et al., 2010), providing a potential early warning for rapid climate change. Isotopomers of $N_2O$ provide information on the sources (Pérez et al., 2000, 2001; Park et al., 2011) i.e. whether $N_2O$ originates predominately from nitrification or denitrification processes. As the conditions/ circumstances leading to emissions from the two processes differ both for the marine and terrestrial sources measuring isotopomers potentially improves our understanding of the climate conditions leading to the release of $N_2O$ over rapid climatic changes.

The $N_2O$ molecule has an asymmetric linear structure (N=N=O) where the position of the $^{15}N$ can be discriminated. The position in the $N_2O$ molecule are named $^{15}N^{\alpha}$ and $^{15}N^{\beta}$ or short $\alpha$ and $\beta$ for $^{14}N^{15}N^{16}O$ and $^{15}N^{14}N^{16}O$, respectively (Yoshida and Toyoda, 2000). The two isotopomers can be distinguished by isotope ratio mass spectrometry, if measurements of the NO fragment are included. A distinction is furthermore possible using mid-infrared spectroscopy because the rotational and vibrational conditions are different for the two isotopomers providing spectral regions where absorptions of the two isotopomers do not overlap (Waechter et al., 2008; Mohn et al., 2010, 2012; Köster et al., 2013b; Heil et al., 2014). For isotopomer measurements at low $N_2O$ concentration (low ppm range), a joint instrument development was executed, applying cavity ring down spectroscopy (CRDS) to enable continuous measurements of the isotopomer abundances and yielding values for $^{15}N^{\alpha}$ and $^{15}N^{\beta}$.

The isotopic composition of a sample is reported as delta values which represents the deviation of the elemental isotope ratio $R_{sample}$ in the sample from a standard $R_{std}$ (Eq. 1). Delta values can be calculated for bulk $N_2O$ as well as for $\delta^{15}N^{\alpha}$ and $\delta^{15}N^{\beta}$. All results are reported relative to the isotopic composition of atmospheric nitrogen.

$$\delta^{15}N = \frac{R_{Sample}}{R_{Std}} - 1 \text{ where } R = \frac{[^{15}N]}{[^{14}N]} \tag{1}$$



The $N_2O$ bulk isotopic composition calculates as the average of $\delta^{15}N^\alpha$ and $\delta^{15}N^\beta$ (Eq. 2) while the site preference (SP) is by definition their difference (Eq. 2) (Brenninkmeijer and Röckmann, 1999; Toyoda et al., 2002).

$$SP = \delta^{15}N^\alpha - \delta^{15}N^\beta \;\;,\;\; \delta^{15}N^{bulk} = \frac{\delta^{15}N^\alpha + \delta^{15}N^\beta}{2} \tag{2}$$

There are multiple natural and anthropogenic sources of $N_2O$. The primary anthropogenic sources of $N_2O$ are organic and inorganic N fertilizers used for agriculture. The natural sources are primarily nitrification and denitrification in terrestrial and aquatic ecosystems (Mosier et al., 1998; Olivier et al., 1998).

Denitrification is a stepwise biological reduction process in which denitrifying bacteria ultimately produce nitrogen ($N_2$). Under anaerobic conditions the denitrifying bacteria use nitrate ($NO_3^-$) instead of oxygen as an electron acceptor in the respiration of organic matter. Through multiple anaerobic reactions $N_2$ is produced as the end product of the complete denitrifying process (reaction R1) (Firestone and Davidson, 1989).

$$NO_3^- \rightarrow NO_2^- \rightarrow NO \rightarrow N_2O \rightarrow N_2 \tag{R1}$$

Each of these anaerobic reactions is carried out by a genuine enzyme, i.e., the production of $N_2O$ is caused by the reaction between nitric oxide (NO) and the enzyme nitric oxide reductase (NOR). The NOR enzyme works as a catalyst in the reduction of NO as shown in reaction R2 (Wrage et al., 2001; Tosha and Shiro, 2013).

$$2NO + 2e^- + 2H^+ \rightarrow N_2O + H_2O \tag{R2}$$

The cleavage of the covalent N=O bond of $N_2O$ leading to $N_2$ and $H_2O$ is the result of $N_2O$ reduction during bacterial denitrification (R3). According to kinetic isotope theory, the cleavage of $N_2O$ is expected to have an increased fractionation effect on $^{15}N^\alpha$, due to a stronger $^{15}N-O$ bond compared to the $^{14}N-O$ (Popp et al., 2002), diffusion into the cell (Tilsner et al., 2003), and enzymatic reduction (Wrage et al., 2004). $N_2O$ reduction during bacterial denitrification is therefore expected to lead to an increase in SP.

$$N_2O + 2e^- + 2H^+ \rightarrow N_2 + H_2O \tag{R3}$$

Reactions with different enzymes typically result in specific isotopic fractionation. The isotopic composition of the intermediately produced $N_2O$ during denitrification is a consequence of multiple reaction steps. Two species of denitrifying bacteria with slightly different enzymes potentially leads to different fractionation. In this study, we compared the fractionation of two contrasting denitrifying bacteria; *Pseudomonas fluorescens* producing and reducing $N_2O$, and *Pseudomonas chlororaphis* producing but not reducing $N_2O$.

## 2 Method

Our objective was to perform continuous position dependent $\delta^{15}N$ measurements of two different bacterial cultures during incubation experiments. Using two denitrifying bacterial cultures we determined the isotopic fractionation and SP during production and reduction of $N_2O$, respectively.



## 2.1 Instrumentation

Bacterial production of $N_2O$ was continuously measured by mid-infrared cavity ringdown spectroscopy using a prototype of the Picarro G5101-i analyzer (in the following named G5101i-CIC) (Picarro, Santa Clara, California, USA). The measurements are non-destructive and are therefore suitable for incubation experiments. The CRDS instrument measures the $^{14}N^{14}N^{16}O$, $^{15}N^\alpha$ and $^{15}N^\beta$ absorption features of $N_2O$ in the wavelength region between $2187.4 \text{ cm}^{-1}$ and $2188 \text{ cm}^{-1}$ (Balslev-Clausen, 2011). The typical precision of the instrument over 10 minutes averaging is $< 0.3$ ppb for the $N_2O$ mixing ratio and $< 0.4$ ‰ for each of the delta values of the isotopomers for concentrations in the range of 200 ppb – 2000 ppb.

Measurements are made by placing the sample delivery system of the G5101i-CIC in a closed loop with a microbial incubation glass chamber (Fig. 1). Circulation is provided by a leak-reduced diaphragm pump installed downstream from the analyzer. The pump (KNF N84.4 ANE) has been sealed using vacuum sealant (Celvaseal high vacuum leak sealant, Myers vacuum repair service, Inc., Kittanning, PA 16201, USA). A low leak rate is prerequisite for accurate measurements in a closed loop experiment. Before the analyzer, a Nafion unit and an Ascarite trap is installed. The Nafion unit removes $H_2O$ vapor whereas the Ascarite trap $(Mg(ClO_4)_2)$ chemically removes $CO_2$. Both gases are removed to exclude potential spectral interference with $N_2O$ in the cavity of the analyzer. Underneath the glass chamber a magnetic stirrer is installed. The stirrer serves two purposes 1) ensure a complete mixing of the bacterial solution and the added nutrient, i.e. potassium nitrate $(KNO_3^-)$, and 2) facilitate gas exchange.

In addition to the measuring mode, the system can be flushed with $N_2$ (not shown in Fig. 1). The flushing mode is used to obtain an anaerobic starting point of the incubation experiment free of $N_2O$. The flushing procedure is fully automated to ensure reproducibility. The entire incubation setup is flushed with $N_2$ for 310 seconds at a high flow rate. The resulting overpressure in the incubator is released prior to switching back to the closed loop position.

## 2.2 Correction of CRDS concentration dependence

Isotopomer measurements made with the G5101i-CIC have a $N_2O$ concentration dependence and need to be corrected. A concentration dependent correction is required because there is a 1/concentration dependence, caused by small offsets in the measurement of the $^{14}N^{15}N^{16}O$ and $^{15}N^{14}N^{16}O$ peaks. These offsets are caused by baseline ripple created by optical cavity etalons. An etalon is an optical effect in which a beam of light undergoes multiple reflections between two reflecting surfaces, and whose resulting optical transmission or reflection is periodic in wavelength. The ripples are not always constant in phase, which means that the ripples can shift spectrally, which can cause the offset to drift over time. The result is a concentration dependent offset to $\delta$ of the form $\pm 1$/concentration. Because baseline ripple effects become more dominant as $N_2O$ concentration decreases, the offset is largest at low concentrations.

Fig. 2 shows results from seven dilution experiments where we gradually mixed a pure $N_2O$ gas with a $N_2/O_2$ mixture (20.1 % $O_2$ and 79.9 % $N_2$, purity 99.999 %). Measurements were performed in a 60 minutes stepwise (18 steps) sequence of both increasing and decreasing concentrations (fig. 2).





We chose to fit the raw data with a cubic spline smoothing function (CSS-function) (Brumback and Rice, 1998). The best fit of these CSS-functions are found (using a smoothing parameter of p = 0.999) in a regression analysis. Four outliers were identified to be outside the $2\sigma$ boundary and were removed from the data set (the red circles in fig. 2). After these outliers were removed the best fit was found again (p = 0.999) and the concentration dependent correction was applied as shown with the

5 green profiles in Fig. 2. Over the course of the experiments, no further instrumental drift was observed.

### 2.3 Calibration gases

Our working standards are CIC-MPI-1 and CIC-MPI-2, prepared based on two standard gases provided by J. Kaiser at University of East Anglia (UEA), Norwich, United Kingdom. These standard gases, MPI-1 and MPI-2, are pure $N_2O$ gases with different isotopic composition (Kaiser, 2002). In our laboratory, each of the standard gases was diluted with a $N_2/O_2$ mixture

(20.1 % $O_2$ and 79.9 % $N_2$, purity 99.999 %) resulting in the two new standard gases, CIC-MPI-1 and CIC-MPI-2. We had both CIC-standard gases measured at four different laboratories to ensure consistent isotopic values (see Table 1). The gases were originally measured at University of East Anglia in Norwich, United Kingdom (UEA). After mixing the gases were measured three times using GC/IRMS measurements at Tokyo Institute of Technology in Japan. At Institute for Marine and Atmospheric research Utrecht in The Netherlands (IMAU) the gases were measured 22 times using GC/IRMS. At the Centre for Ice and

Climate in Copenhagen Denmark (CIC) the gases were measured continuously over two hours. All measurements are reported relative to atmospheric $N_2$. The root-mean-square deviations (RMSD) calculated from the three laboratories with respect to the original UAE data (Table 1) are similar to the RMSD calculated by Mohn et al. (2014). They present RMSD calculations based on gas measurements performed at 12 different laboratories using both IRMS and laser spectroscopy.

### 2.4 Pure bacterial cultures

The two bacterial cultures used in this study are both gram-negative bacteria with the capability to denitrify, i.e. reduce nitrate to gaseous nitrogen. Isolates were obtained from an agricultural soil of sandy loam type (Roskilde Experimental Station) on 11 April 1983. One culture is a *Pseudomonas fluorescens*, bio-type D that reduces $NO_3^-$ all the way to $N_2$. The second culture, *Pseudomonas chlororaphis*, is only capable of reducing $NO_3^-$ to $N_2O$ (Christensen and Bonde, 1985), which means that the nitrous oxide reductase is absent or at least not active in this organism. The latter bacterium is contained in the American Type

Culture Collection with accession number ATCC 43928 (Christensen and Tiedje, 1988). The cultures were grown anoxic in 50 ml serum bottles with 1/10 tryptic soy broth (Difco) added 0.1 g $KNO_3 \cdot L^{-1}$. After six days of growth at room temperature (24°C), *P. chlororaphis* had converted all N in $NO_3^-$ into $N_2O$. The bacterial culture of *P. Fluorescens* was cultivated for six days at a slightly lower temperature (15°C) to assure that the cultures were in a comparable phase of potential activity when assayed for gas production/reduction activity. The six days old cultures were used in the incubation experiment where it is

conditional for the denitrifying process that organic carbon is available, that the concentration of oxygen is low, and that the concentration of $NO_3^-$ is high (Wrage et al., 2001; Stuart Chapin III et al., 2002).



### 2.4.1 Bacterial incubation experiments

50 mL bacterial solution of *P. chlororaphis* or *P. fluorescens* was placed in a petri-dish in the 1000 mL incubation chamber. Hereafter, the setup was flushed with pure $N_2$ (purity 99.9999 %) to ensure anaerobic conditions. To ensure no $N_2O$ gas exchange prior to the experiment, the bacterial solution was left for 90 minutes under constant magnetic stirring. Then the

incubation chamber was opened and the bacterial solution was fed with 2.5 mL and 15 mL 0.45 mM $KNO_3$ for *P. chlororaphis* and *P. fluorescens*, respectively. The incubation experiment started by again flushing the setup with pure $N_2$ immediately after the addition of $KNO_3$.

A total of seven replicate incubations of the full denitrifying bacteria (*P. fluorescens*) and five replicate incubations of the denitrifying bacteria with no active nitrous oxide reductase (*P. chlororaphis*) were executed. All of the cultures were

continuously measured from the moment $KNO_3$ was added to the bacterial incubations. We terminated experiments with *P. fluorescens* when the $N_2O$ concentration was below 0.2 ppm. For *P. chlororaphis*, we defined the end of the experiment when the $N_2O$ concentration had reached a constant level for 200 minutes.

Continuous measurements of the bacterial production of $N_2O$ from *P. chlororaphis* were performed for approximately 500 minutes for each replica. All five replicas were measured within one week, starting at the same hour of the day and after equally

long cultivation prior to the measurements. The bacterial evolution, of $N_2O$ production and reduction, from *P. fluorescens* was continuously measured for about 1000 minutes. The seven replicas were measured during three one week measuring campaigns over the course of half a year, always with an equally long cultivation prior to measurements.

Our CIC-MPI standards contain $N_2O$ in an $O_2/N_2$ mixture while our incubation measurements were carried out in an $N_2$ atmosphere. This difference has a significant effect on the spectroscopy. We find a linear dependence similar to the one found

by (Erler et al., 2015). The presented data have been corrected for the effect, which is 15.27 ‰, 11.66 ‰, and 13.47 ‰ for $\delta^{15}N^{\alpha}$, $\delta^{15}N^{\beta}$, and $\delta^{15}N^{bulk}$ respectively (see supplementary). Given the leak rate of the system we estimate that the $O_2$ concentration has not increased by more that 0.002 % over the course of the experiment. We therefore neglect the effect of a continuously increasing $O_2$ concentration and assume a constant $O_2$ concentration of 0 %.

### 2.5 Determination of isotopic fractionation

Our incubation experiments are Rayleigh type experiments. In the following, we determine the underlying isotope fractionation from the isotopic evolutions in the incubator under this assumption. Rayleigh fractionation describes the changing isotopic composition in the substrate and product of a unidirectional one-step reaction. Lord Rayleigh (1896) derived the equation for the fractionating isotope ratio of the substrate (Eq. 3) as:

$$\frac{R_s}{R_{s,0}} = f^{\left(\alpha_{p/s}-1\right)} \tag{3}$$

where $R_{s,0}$ is the initial isotope ratio of the substrate, $R_s$ is the isotope ratio of the substrate at time $t$, $\alpha_{p/s}$ is the fractionation factor of the product versus the substrate, and $f$ is the unreacted fraction of substrate at time $t$. The corresponding equation for



the isotopic ratio of the accumulated product $R_{p,acc}$ derives as given in Eq. 4.

$$R_{p,acc} = R_{s,0} \cdot \left( \frac{1 - f^{\alpha_{p/s}}}{1 - f} \right) \qquad (4)$$

The isotopic fractionation is defined as $\epsilon = \alpha - 1$. We do not know the isotopic composition of $KNO_3$ used for our experiments. However, when all $KNO_3$ has reacted to $N_2O$, its isotopic composition is identical to the initial composition of $KNO_3$. The

5 final isotopic values of $N_2O$ for *P. chlororaphis* can therefore be used to estimate the initial composition of $KNO_3$. We therefore get the initial isotopic composition of $KNO_3$ to be -5.03 ‰ ± 0.79 for $\delta^{15}N^{\alpha}$, -0.89 ‰ ± 2.36 for $\delta^{15}N^{\beta}$, and -3.08 ‰ ± 1.05 for $\delta^{15}N^{bulk}$.

### 2.5.1 Fractionation associated with N$_2$O production

The fractionation associated with production of $N_2O$ is the result of multiple reactions with various isotopic fractionations (Ostrom and Ostrom, 2011). Although, Ostrom and Ostrom (2011) argue that the fractionation observed during production of $N_2O$ via reduction of $NO_3^-$ can be treated as a net isotope effect with $NO_3^-$ being the substrate and $N_2O$ as the product. During production of $N_2O$ Eq. 4 is therefore used in calculation of the net fractionation factor and the net isotope effect of $\delta^{15}N^{bulk}$. For the two isotopomers Eq. 4 needs to be slightly modified (see supplementary). The SP isotopic fractionation is derived using

the relationship $\epsilon_{sp} = \epsilon_{\alpha} - \epsilon_{\beta}$ as stated by Ostrom et al. (2007) and Well and Flessa (2009a).

### 2.5.2 Modifications to the Rayleigh model

To describe the evolution of the isotopomers during production of $N_2O$, the Rayleigh process is not directly applicable to the isotopomers as the two isotopomers both are direct products of the same denitrification process from the same batch of denitrifying bacteria and nitrate. The accumulated product for the two isotopomers can be determined (as presented in the

20 supplementary). The calculations results in an isotopomer correction factor $\varphi_{\alpha}$ and $\varphi_{\beta}$ for the two isotopomers respectively.

$$\varphi_{\alpha} = \frac{\alpha_{\alpha}}{\alpha_{bulk}} \qquad (5)$$

$$\varphi_{\beta} = \frac{\alpha_{\beta}}{\alpha_{bulk}} \qquad (6)$$

The fractionation factor for the bulk is denoted $\alpha_{bulk}$, while the fractionation factors for the two isotopomers are presented as

$\alpha_{\alpha}$ and $\alpha_{\beta}$, respectively. The Rayleigh equation for the accumulated product of $\delta^{15}N^{\alpha}$ ($R_{p,acc}^{\alpha}$) is therefore as presented in Eq. 7, in which $R_{p,acc}^{bulk}$ is the accumulated product calculated for $\delta^{15}N^{bulk}$. A similar equation exists for $R_{p,acc}^{\beta}$, using $\delta^{15}N^{\beta}$ and $\varphi_{\beta}$:

$$R_{p,acc}^{\alpha} = R_{p,acc}^{bulk} \cdot \varphi_{\alpha} \qquad (7)$$





Equation 7 is valid for *P. chlororaphis*. For *P. fluorescens*, both an immediate reduction and an uptake reduction take place simultaneously with $N_2O$ production due to the pre-experimental cultivation leading to activation of all enzymes in the bacterial solution. Part of the freshly produced $N_2O$ is therefore immediately reduced to $N_2$. This reduction is fractionating with fractionation factor $\alpha_\alpha^R$, $\alpha_\beta^R$, and $\alpha_{bulk}^R$, respectively for each of the isotopomers and the bulk. The isotope imprint of the re-

duction on the remaining $N_2O$ depends on the ratio between reduction rate and production rate, from here on referred to as the reduction correction parameter ($\gamma$). Assuming $\gamma$ is constant (for one experiment cf. Fig. 3B) results in the following first order approximation of the accumulated product including the isotope imprint of the reduction on the remaining $N_2O$ ($R_{p,r}^\alpha$).

$$R_{p,r}^\alpha = \frac{R_{p,acc}^\alpha \cdot \left(1 - \alpha_\alpha^R \cdot \gamma\right)}{1 - \gamma} \tag{8}$$

Equation 8 is presented for $\delta^{15}N^\alpha$, though similar equations is used for both $\delta^{15}N^\beta$ and $\delta^{15}N^{bulk}$. For any calculated ratio the

values are given in ‰ using the delta-notation (Eq. 1).

### 2.5.3    Isotopic fractionation associated with $N_2O$ reduction

The isotopic fractionation during reduction of $N_2O$ to $N_2$ derives from Eq. 3. Equation 3 is valid for the bulk $^{15}N$ / $^{14}N$ isotope ratios of $N_2O$ ($\delta^{15}N^{bulk}$) and for both of the two isotopomer isotope ratios ($\delta^{15}N^\alpha$ and $\delta^{15}N^\beta$) (Mariotti et al., 1981; Menyailo and Hungate, 2006; Ostrom et al., 2007; Lewicka-Szczebak et al., 2014).

### 2.5.4    Fitting procedure

We determine the respective isotopic fractionation during production and reduction of $N_2O$ for each of the bacterial strains assuming a Rayleigh type process. The best fit (highest $R^2$) is found using an iterative approach between the measured data and the Rayleigh fractionation model for the accumulated product.

For *P. chlororaphis* (being a pure producer of $N_2O$) we use Eq. 4 and 7 as the Rayleigh fractionation model for the $\delta^{15}N^{bulk}$ and the two isotopomers, respectively. We fit the Rayleigh fractionation model to the measured data. The highest $R^2$ values are iteratively found, using iterative determination of the fractionation factor during production of $N_2O$ ($0 \leq \alpha \leq 1$). The limits of the unreacted fraction of the substrate ($f$) was set to $f_{start}$ = 1 and $f_{end}$ = 0.

Since $N_2O$ is the product and the substrate for production and reduction respectively Eq. 8 and 3 are applied for *P. fluorescens* to the corresponding process. We defined the section of production as being from the start of the measurements until the net production (net emission) rate turns negative. From the calculations of the net production rates (see Fig. 3) we believe that $N_2O$ production continues after the point of maximum concentration. However, at one point $NO_3^-$, $NO_2^-$ and NO are fully consumed and *P. fluorescens* is forced to exclusively reduce $N_2O$. We defined the start of the section where *P. fluorescens* is only reducing

$N_2O$ to the point where both $\delta^{15}N^\alpha$ and $\delta^{15}N^\beta$ start decreasing (assumption based on reduction of $\delta^{15}N^\alpha$, $\delta^{15}N^\beta$, $\delta^{15}N^{bulk}$, and concentration). Between the end of the net production and the start of the exclusive reduction, no Rayleigh model can be fitted.



The fractionation factors resulting in the highest $R^2$ values are picked as the best fit fractionation factor for each specific evolution. In the calculations of the $N_2O$ production from *P. fluorescens* the fractionation factor during $N_2O$ reduction is required. We therefore first fit the Rayleigh fractionation model (Eq. 3) to the measured $N_2O$ reduction data. The highest $R^2$ values are iteratively found, using 1) the extremes of the unreacted fraction of the substrate being $f_{start} = 1$ and $f_{end} = 0$, 2)

the fractionation factor during reduction of $N_2O$ ($1 \leq \alpha \leq 2$). Secondly we fitted the Rayleigh fractionation model (Eq. 8) to the measured $N_2O$ production data. The highest $R^2$ values are iteratively found by varying 1) the extremes of the unreacted fraction of the substrate ($f_{start}$ and $f_{end}$) with $1 \leq f_{start} < f_{end} \leq 0$ as the boundary conditions, 2) the reduction correction parameter ($\gamma$) in the range $0 \leq \gamma \leq 1$, and 3) the fractionation factor during production of $N_2O$ ($0 < \alpha < 1$), and using the fractionation factor during $N_2O$ reduction (from calculations above).

Nitric oxide (NO) accumulation in the headspace can not be excluded from any of the presented $N_2O$ production experiments. The effect of accumulation of NO in the headspace is assumed to be addressed by iterative determination of $f_{end}$ and $f_{start}$ (Table 4 and Table 2 for *P. chlororaphis* and for *P. fluorescens*, respectively), hence no further adjustment is taken.

## 3   Results

The evolution of $N_2O$ concentration over time from the two bacterial strains shows two very distinctive patterns with both an

increasing and decreasing $N_2O$ concentration characteristic for *P. fluorescens* and an increasing $N_2O$ concentration followed by a stabilization characteristic for *P. chlororaphis* (Fig. 3A) as has previously been described by Christensen and Tiedje (1988). These distinctive characteristics are only vaguely seen in the respective dynamics of the SP for the two bacterial strains (Fig. 4A and 4B).

### 3.1   Pseudomonas chlororaphis

Paralleling the increase in $N_2O$ concentration (Fig. 3A), we also find an increase in $\delta^{15}N^\alpha$, $\delta^{15}N^\beta$ and $\delta^{15}N^{bulk}$ over time (Fig. 5A and 5B). The final product of *P. chlororaphis* is $N_2O$; this is a unidirectional transfer of nitrogen from $KNO_3$ to $N_2O$ and thereby a Rayleigh process although multiple fractionations are involved. Figure 5A and 5B shows the best fit Rayleigh profile for $\delta^{15}N^\alpha$ and $\delta^{15}N^\beta$ respectively.

The modeled Rayleigh distillation profiles were found to match the production of $N_2O$ from *P. chlororaphis* to a relatively

high degree. The average coefficient of determination ($R^2$) between data and fitted Rayleigh curves are 72.15 %, 64.20 %, and 76.31 % for $\delta^{15}N^\alpha$, $\delta^{15}N^\beta$ and $\delta^{15}N^{bulk}$ respectively. The calculations of the isotopic fractionation for the fractionation of $\delta^{15}N^\alpha$ give a mean value of -6.72 ‰ $\pm$ 1.54 (Table 4). For $\delta^{15}N^\beta$ the mean isotopic fractionation was found to be -3.28 ‰ $\pm$ 1.37 (Table 4). These values leads to a mean SP isotopic fractionation value of -3.42 ‰ $\pm$ 1.69 and a $\delta^{15}N^{bulk}$ isotopic fractionation of -5.01 ‰ $\pm$ 1.20 (Table 4).





### 3.2  Pseudomonas fluorescens

Continuous measurements of the evolution of $N_2O$ produced and consumed by the denitrifying bacteria *P. fluorescens* are presented in Fig. 6A and 6B for $\delta^{15}N^\alpha$ and $\delta^{15}N^\beta$, respectively. The correlation coefficient of the fitted Rayleigh model for the production matches the continuously measured $\delta^{15}N^\alpha$ data by 94.44 % on average using the $R^2$ method for the seven

replicates of *P. fluorescens* incubations. Equivalent $R^2$ average for $\delta^{15}N^\beta$ are 94.13 %, whereas the average for $\delta^{15}N^{bulk}$ are found to be 96.86 %. The $R^2$ found for the reduction part is 91.92 % for $\delta^{15}N^\alpha$, 81.09 % for $\delta^{15}N^\beta$, and 92.42 % for $\delta^{15}N^{bulk}$ on average for the seven replicates. The fractionation during both the production and the reduction are therefore following the Rayleigh fractionation model to a large degree. The isotopic fractionation calculated using these models are therefore a good representation for the fractionation caused by the *P. fluorescens* bacteria on the $N_2O$. The resulting isotopic fractionation

is presented in Table 2 for the production part and in Table 3 for the reduction part together with the calculated isotopic fractionation for the SP. During production of $N_2O$, the mean isotopic fractionation for SP was found to be $\epsilon_{SP} = 5.73\ \permil \pm$ 5.26 while the $\epsilon_{bulk} = $ -52.21 $\permil \pm$ 9.28 for the $\delta^{15}N^{bulk}$, hence there is a difference of 9.15 $\permil$ and 47.20 $\permil$ for $\epsilon_{SP}$ and $\epsilon_{bulk}$, respectively, to *P. chlororaphis*. During reduction of $N_2O$ the mean isotopic fractionation for SP was found to be $\epsilon_{SP} = 2.41$ $\permil \pm$ 3.04 and $\epsilon_{bulk} = 8.77\ \permil \pm$ 4.49 for $\delta^{15}N^{bulk}$.

## 4   Discussion

The two bacteria investigated are denitrifiers, in principle functionally similar but with *P. chlororaphis* lacking the ability to reduce $N_2O$ to $N_2$. Both denitrifiers were cultivated under anaerobic conditions leading to active nitric oxide reductase (both cultures) and nitrous oxide reductase (*P. fluorescens*). Fed the same amount of nitrate, we therefore expect the maximum $N_2O$ concentration and the net $N_2O$ production rate to be lower for *P. fluorescens* than for *P. chlororaphis*.

In Fig. 3, an example of the $N_2O$ evolution by the two bacterial strains is plotted. The net $N_2O$ production by *P. chlororaphis* is higher and a higher level in concentration is reached than by *P. fluorescens* even though *P. fluorescens* received six times more substrate than *P. chlororaphis*. These observations indicate that the nitrous oxide reductase consumes $N_2O$ from a very early stage of the $N_2O$ turnover, likely because the cultures were grown under anaerobic conditions leaving both $N_2O$-producing and -consuming enzymes active from the beginning of the experiment.

$N_2O$ produced by the two denitrifying bacteria differs primarily in the bulk isotopic fractionation whereas the SP isotopic fractionation averages at a more comparable level. We find that the average difference in isotopic fractionation between *P. chlororaphis* and *P. fluorescens* is 56.58 $\permil$ and 10 $\permil$ for $\epsilon_{bulk}$ and $\epsilon_{SP}$, respectively (Fig. 7). I.e. the isotopic fractionation is significantly higher for both isotopomers in *P. fluorescens* than in *P. chlororaphis*.

The observed difference in bulk isotopic fractionation during production of $N_2O$ could originate from (1) a difference in the production rate, (2) a difference in the nitric oxide reductase enzymes, (3) a difference in nitrite reductase, or (4) fractionation caused by nitrous oxide reductase.



(1) The production rates in our experiments (Table 2 and 4) show a correlation between isotopic fractionation and production rate similar to the one observed by Mariotti et al. (1982). In our experiments the production rate for *P. chlororaphis* is 10 times higher than for *P. fluorescens*, which cf. Mariotti et al. (1982) would account for only approximately a 10 ‰ offset. We therefore believe that a change in production rate does not account for the 56.58 ‰ difference in bulk isotopic fractionation alone.

(2) Nitric oxide reductase is the primary enzyme in a chain of catalytic reactions leading to the production of $N_2O$ (Hino et al., 2010; Hendriks et al., 2000). The catalytic cycle involving production of $N_2O$ from NO has yet to be completely understood with respect to the formation of the N-N double bond, the complexity of the structural information of nitric oxide reductase, the proton transfer pathway into nitric oxide reductase (Tosha and Shiro, 2013), and the very short lifetime of the intermediate states of the molecules (Collman et al., 2008). We hypothesized that the difference in the bulk observed during

incubation of our two bacterial species was due to different nitric oxide reductases produced by the two species. To test this hypothesis, we compared the DNA sequences of the norB and norC genes coding for the large and small subunit of nitric oxide reductase, respectively, from three different strains of *P. fluorescens* (strains NCIMB 11764 [Genbank accession number CP010945], PA3G8 [742825335], and F113 [CP003150]) and *P. chlororaphis* (strains O6 [389686655], PA23 [749309655], and UFB2 [836582503]) as well as two closely related denitrifying species, *P. aeruginosa* and *P. stutzeri*. Our analysis revealed

1) a very high similarity of the two genes in *P. fluorescens* and *P. chlororaphis* and 2) that the intra-species variability of the two genes was similar to the inter-species variation. This led us to reject our hypothesis and we conclude that differences in nitric oxide enzymes produced by the two species were not responsible for the observed differences in the bulk isotopic fractionation.

    (3) Nitrite reductase lead to the first gaseous product during the denitrification pathway and it has been found to play an important role in isotope fractionation (Martin and Casciotti, 2016). The DNA sequences of the two bacterial strains used

in our experiments revealed *P. chlororaphis* having a copper-containing nitrite reductase encoded by the gene nirK while *P. fluorescens* has a heme containing cytochrome cd1-type nitrite reductase encoded by the gene nirS. The two enzymes are structurally very different and show significantly different isotope fractionations on $\epsilon_{bulk}$ (Martin and Casciotti, 2016). Sutka and Ostrom (2006) conclude that a difference in the nitrite reductase does not have an effect on the SP during denitrification. This conclusion is based on measurements of two denitrifiers (*P. chlororaphis* (ATCC 43928) and *P. aureofaciens* (ATCC

13985)) possessing cd1-type nitrite reductase and Cu-containing nitrite reductase, respectively. We conclude the same for the two denitrifying cultures presented in this work (*P. fluorescens* and *P. chlororaphis*) since the difference in isotopic fractionation between the two bacteria species primarily is on $\epsilon_{bulk}$.

    (4) The experiments involving *P. fluorescens* are hypothesized to experience an increased fractionation during both production and reduction of $N_2O$ as a result of simultaneous production and reduction of $N_2O$. After production of $N_2O$ molecules,

the light molecules degas quickly out of the liquid phase and away from the nitrous oxide reductase. The heavy $N_2O$ molecules are slower and react with the nitrous oxide reductase leading to a depletion of the heavy $N_2O$ isotopes. As this is a diffusion driven process it is mass-dependent and given the difference in bond energy required for breakage of the N-O bond a slight effect on SP is expected. This is also in line with the kinetic isotope effect theory which suggest a small offset towards higher $\epsilon_\alpha$ and therefore higher $\epsilon_{SP}$ (Popp et al., 2002; Tilsner et al., 2003; Wrage et al., 2004; Well and Flessa, 2008).





We hypothesized that the difference in $\epsilon_{bulk}$ between the two bacteria originates from 1) a difference in production rates, 3) a difference in fractionation caused by different nitrite reductases, and 4) fractionation associated with nitrous oxide reductase in *P. fluorescens*. The slight difference in $\epsilon_{SP}$ is hypothesized to stem from 1) a difference in net production rates, and 4) fractionation associated with nitrous oxide reductase in *P. fluorescens*.

## 4.1 Comparison to previous studies

Numerous publications have presented experiments with both in situ measurements of denitrifying bacterial production and reduction of $N_2O$ during incubation of bacterial cultures and soil samples. In Fig. 7, we present a comparison between the results from this study and the results from a selection of the previously published results. The general understanding is that denitrification results in SP $\leq$ 9 ‰. Applying different incubation techniques on soils with different properties showed SP during production of $N_2O$ between -3 ‰ and 9 ‰ and between -0.9 ‰ and -8.2 ‰ during reduction (Well and Flessa, 2009a, b; Köster et al., 2013a; Lewicka-Szczebak et al., 2014, 2015). During production of $N_2O$ in bacterial culture experiments involving *P. chlororaphis* (ATCC 43928), *P. aureofaciens* (ATCC 13985), and *P. denitrificans* (ATCC 17741), Thompson et al. (2004); Toyoda et al. (2005); Sutka and Ostrom (2006); Ostrom and Ostrom (2011) found $\epsilon_{SP}$ values of -27.4 ‰ and 1.3 ‰ and $\epsilon_{bulk}$ values of between -42.6 ‰ and 36.7 ‰. The wide range of $\epsilon_{bulk}$ presented both in this study and by others, confirms the difficulties in the use of $\epsilon_{bulk}$ in $N_2O$ evolution analysis. Toyoda et al. (2005) present contrasting results for $\epsilon_{SP}$ of *P. fluorescens* (ATCC 13525) of 24.1 ‰. The result may, however, not be comparable to ours as Toyoda et al. (2005) suspect an abiological reaction within the incubation flask to be responsible for $N_2O$ production in the incubation experiment. Our results for $\epsilon_{SP}$ calculated during $N_2O$ production from both *P. chlororaphis* and *P. fluorescens* are in the same range as what has been reported previously.

A number of studies have investigated $N_2O$ reduction from denitrification in soils (e.g. (Well and Flessa, 2009b; Köster et al., 2013a; Lewicka-Szczebak et al., 2014, 2015)). The results are only partly in accord with our findings for specific bacteria strains. I.e. they find consistently negative $\epsilon_{SP}$ and $\epsilon_{bulk}$ while we find slightly positive values on average. Ostrom et al. (2007) investigated bacterial reduction of $N_2O$ using *P. stutzeri* and *P. denitrificans*, and the $\epsilon_{SP}$ resulting from this bacterial reduction of $N_2O$ was between -6.8 ‰ and -5 ‰. We hypothesize this difference to be due to diffusive isotopic fractionation and a difference in active enzymes during production of $N_2O$.

## 5 Conclusions

We have presented successful continuous measurements of the denitrifying bacterial process using two different strains of bacteria: *P. fluorescens* which is a full denitrifier, and *P. chlororaphis* which is a denitrifier without nitrous oxide reductase activity. Assuming a Rayleigh type fractionation, modified for isotopomers and simultaneous reduction, we have calculated the isotopic fractionation during production and reduction of $N_2O$. The isotopic fractionation for *P. chlororaphis* is in line with previous results for both SP and bulk. For *P. fluorescens*, we find similar SP isotopic fractionation values during $N_2O$ production, while



we find slightly increased SP isotopic fractionation values during $N_2O$ reduction. The bulk isotopic fractionation calculated for $N_2O$ reduction is in line with previously presented results though for production we find an isotope depletion. We believe that, in our experiment, the bulk isotope depletion is due to a difference in production rates, in nitrite reductase, and in nitrous oxide reductase.

5   *Author contributions.*  MW and TB designed the experiments and MW carried out the measurements and analyzed data. SC prepared the bacteria prior to experiments. AP analyzed the bacterial DNA sequences. DBH and EC developed the G5101i-CIC analyzer. MW prepared the manuscript with contribution from all co-authors.

*Acknowledgements.*  We thank Jan Kaiser for isotope specific gas samples used for our reference gases, Sakae Toyoda, Naohiro Yoshida, Carina van der Veen, and Thomas Röckmann for assistance on measurements of our reference gases. We want to thank Center for Permafrost
10  (CENPERM DNRF100) and the Centre for Ice and Climate, funded by the Danish National Research Foundation for their support, and The Danish Agency for Science Technology and Innovation, for funding used in supporting this project. We want to thank Picarro Inc. and especially Eric Crosson, and Nabil Saad for the collaboration and guidance in the development of the $N_2O$ isotope analyzer prototype used in this project.





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





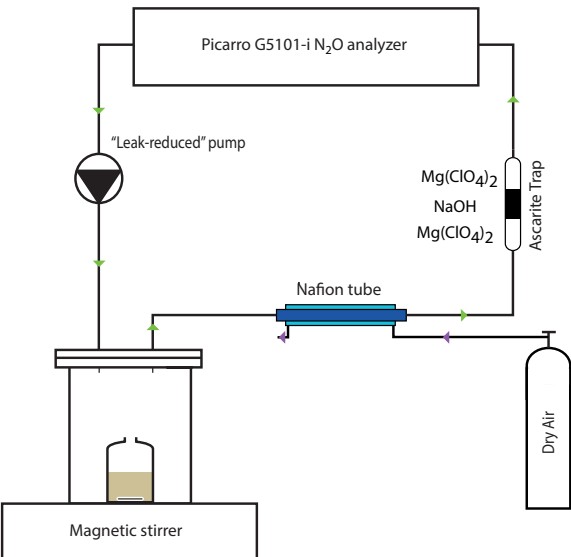

**Figure 1.** Simplified schematic of the incubation setup. The green and the purple arrows show the flow direction of the measuring gas and the purge gas, respectively.

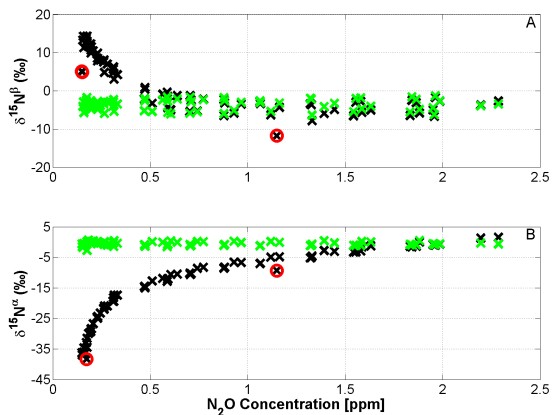

**Figure 2.** The concentration-dependent correction (CDC) for (A) $\delta^{15}N^{\alpha}$ and (B) $\delta^{15}N^{\beta}$ respectively. In both Figures, the raw data are presented in black, the CDC data is plotted in green, and the outliers are marked with red circles.



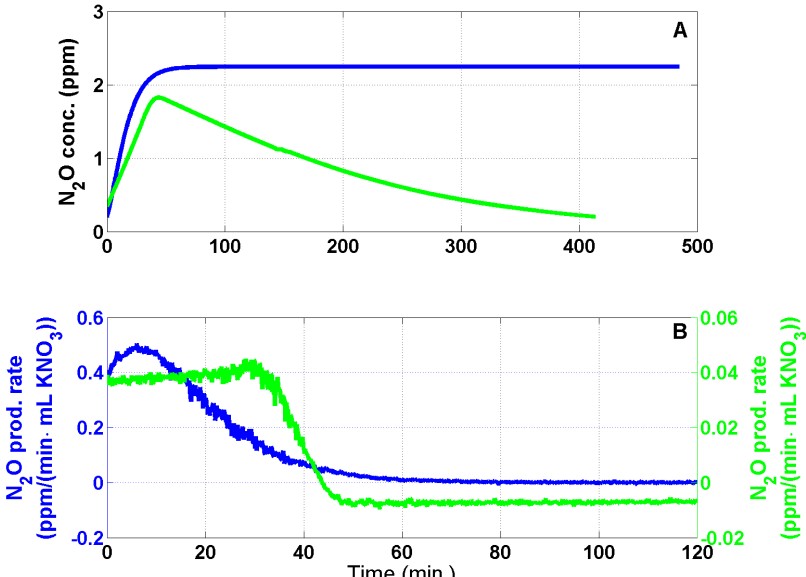

**Figure 3.** Continuous measurements of (A) $N_2O$ concentrations and (B) the net $N_2O$ production rate from experiments with *P. fluorescens* (green) and *P. chlororaphis* (blue), respectively. Only the first 120 minutes of the net $N_2O$ production are shown. Note that the scaling of the two horizontal axis differs.

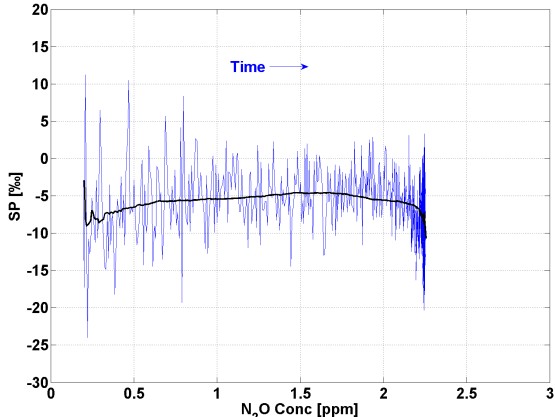

**Figure 4A.** SP as a function of $N_2O$ concentration as produced by *P. chlororaphis*. High resolution CRDS data (blue line) and five minutes running average (black line). The blue arrow indicates the direction of time during production of $N_2O$.



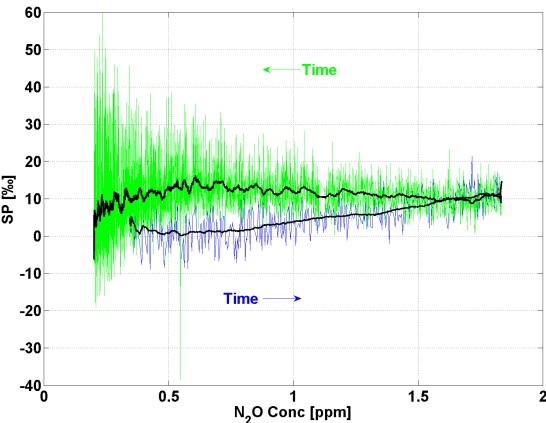

**Figure 4B.** SP as a function of $N_2O$ concentration produced by *P. fluorescens*. High resolution CRDS data (blue and green line) and five minutes running average (black line). The blue arrow indicates the direction of time during production of $N_2O$ whereas the green arrow indicates the direction of time during reduction of $N_2O$.

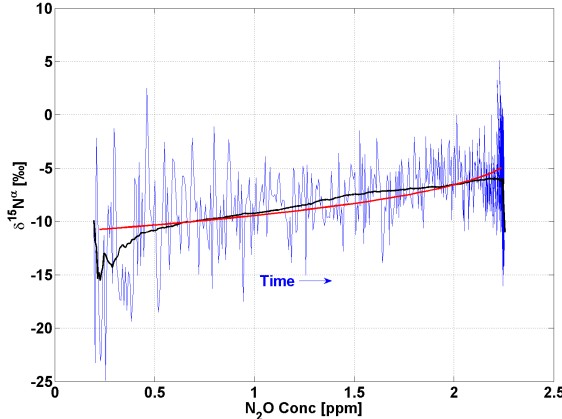

**Figure 5A.** $\delta^{15}N^{\alpha}$ as a function of $N_2O$ concentration as produced by *P. chlororaphis* and the modeled Rayleigh type distillation. High resolution CRDS data (blue line) and five minutes running average (black line). The red curve is the modeled Rayleigh type distillation curve for the production of $N_2O$. The blue arrow indicates the direction of time during production of $N_2O$.




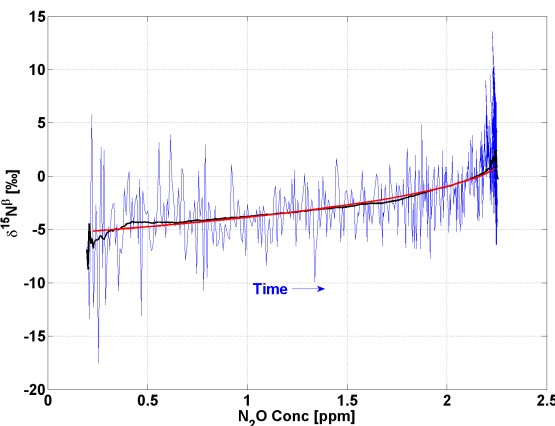

**Figure 5B.** $\delta^{15}N^{\beta}$ as a function of $N_2O$ concentration as produced by *P. chlororaphis* and the modeled Rayleigh type distillation. High resolution CRDS data (blue line) and five minutes running average (black line). The red curve is the modeled Rayleigh type distillation curve for the production of $N_2O$. The blue arrow indicates the direction of time during production of $N_2O$.

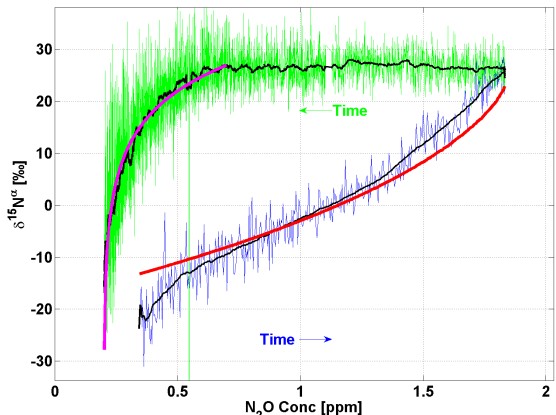

**Figure 6A.** $\delta^{15}N^{\alpha}$ as a function of $N_2O$ concentration produced by *P. fluorescens* with the modeled Rayleigh type distillation. High resolution CRDS data (blue and green line) and five minutes running average (black line). The red and magenta curves are the modeled Rayleigh type distillation curves for the production and reduction of $N_2O$, respectively. The blue arrow indicates the direction of time during production of $N_2O$ whereas the green arrow indicates the direction of time during reduction of $N_2O$.




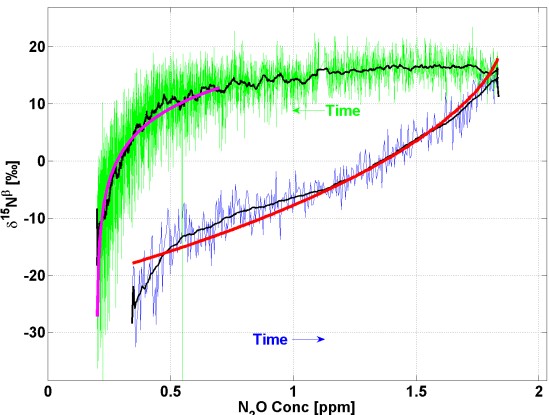

**Figure 6B.** $\delta^{15}N^{\beta}$ as a function of $N_2O$ concentration produced by *P. fluorescens* with the modeled Rayleigh type distillation. High resolution CRDS data (blue and green line) and five minutes running average (black line). The red and magenta curves are the modeled Rayleigh type distillation curves for the production and reduction of $N_2O$, respectively. The blue arrow indicates the direction of time during production of $N_2O$ whereas the green arrow indicates the direction of time during reduction of $N_2O$.

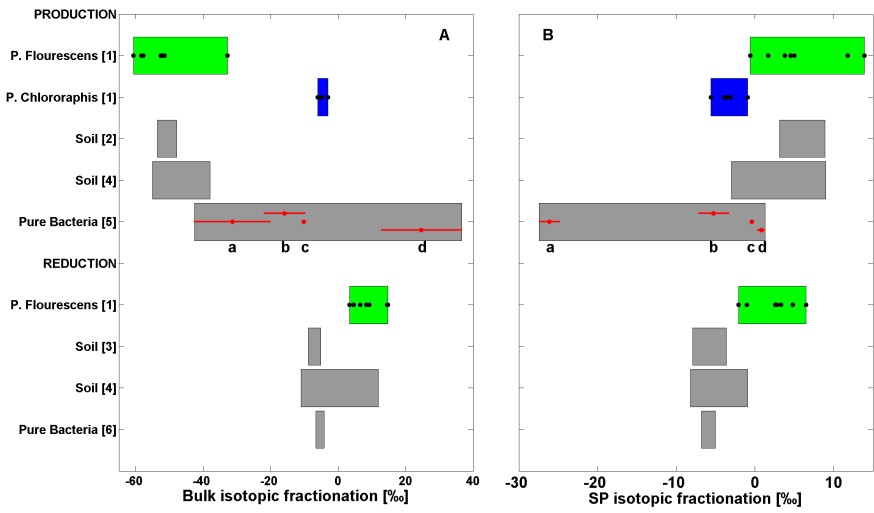

**Figure 7.** (A) Bulk and (B) SP isotopic fractionation calculated from the continuous measurements with *P. fluorescens* (green) and *P. chloraphis* (blue) respectively. Both the production and reduction isotopic fractionations are shown and compared with previously presented results. [1] [This study], [2] [Well and Flessa (2009a)], [3] [Well and Flessa (2009b)], [4] [Lewicka-Szczebak et al. (2014, 2015)], [5] [a) Thompson et al. (2004), b) Toyoda et al. (2005), c) Ostrom and Ostrom (2011) d) Sutka and Ostrom (2006)], [6] [Ostrom et al. (2007)].





**Table 1.** Measurements of standard gas CIC-MPI-1 and CIC-MPI-2, diluted MPI-1 and MPI-2 gases respectively. Measurements were measured at the four respective institutes. The combined mean-values and standard deviations are calculated from measurements performed at Tokyo-Tech, IMAU, and CIC. The root-mean-square deviations (RMSD) are calculated against the original UEA data.

| Std. gas | Institute | $[N_2O]$ (ppb) | $\delta^{15}N^{bulk}$ (‰) | $\delta^{15}N^{\alpha}$ (‰) | $\delta^{15}N^{\beta}$ (‰) |
|---|---|---|---|---|---|
| MPI-1 | UEA | - | $1.00 \pm 0.03$ | $0.70 \pm 0.90$ | $1.30 \pm 0.90$ |
| CIC-MPI-1 | | | | | |
| | Tokyo-Tech | $1828.9 \pm 4.9$ | $1.34 \pm 0.17$ | $1.44 \pm 0.09$ | $1.24 \pm 0.35$ |
| | IMAU | $1919.3 \pm 21.0$ | $1.08 \pm 0.10$ | $2.31 \pm 0.25$ | $-0.15 \pm 0.33$ |
| | CIC | $1918.4 \pm 2.3$ | $2.62 \pm 1.77$ | $0.29 \pm 2.44$ | $4.95 \pm 2.37$ |
| | Mean | $1909.8 \pm 29.3$ | $1.32 \pm 0.82$ | $1.94 \pm 0.87$ | $0.70 \pm 1.42$ |
| | RMSD | - | 0.96 | 1.05 | 2.27 |
| MPI-2 | UEA | | $-1.78 \pm 0.03$ | $12.40 \pm 0.04$ | $-15.90 \pm 0.90$ |
| CIC-MPI-2 | | | | | |
| | Tokyo-Tech | $1840.2 \pm 32.1$ | $-1.30 \pm 0.06$ | $12.79 \pm 0.22$ | $-15.41 \pm 0.24$ |
| | IMAU | $1865.9 \pm 16.0$ | $-1.68 \pm 0.10$ | $11.80 \pm 0.37$ | $-15.16 \pm 0.46$ |
| | CIC | $1827.2 \pm 1.8$ | $-0.15 \pm 1.78$ | $12.58 \pm 2.51$ | $-12.89 \pm 2.35$ |
| | Mean | $1857.9 \pm 19.2$ | $-1.43 \pm 0.82$ | $12.01 \pm 1.06$ | $-14.87 \pm 1.13$ |
| | RMSD | - | 0.98 | 0.43 | 1.81 |

**Table 2.** Isotopic fractionation, reduction correction parameter ($\gamma$), extremes of the unreacted fraction parameter ($f_{start}$ and $f_{end}$), and production rate ($k_p$) in $[ppm/(min \cdot mL\ KNO_3)]$ for the production of $N_2O$ from *P. fluorescens*.

| Replica # | $\epsilon_{\alpha}$ (‰) | $\epsilon_{\beta}$ (‰) | $\epsilon_{bulk}$ (‰) | $\epsilon_{SP}$ (‰) | $\gamma$ | $f_{start}$ | $f_{end}$ | $k_p$ |
|---|---|---|---|---|---|---|---|---|
| 1 | -55.50 | -60.00 | -57.75 | 4.50 | 0.00 | 0.38 | 0.000 | 0.035 |
| 2 | -33.10 | -32.50 | -32.80 | -0.60 | 0.52 | 0.94 | 0.000 | 0.078 |
| 3 | -49.40 | -54.40 | -51.90 | 5.00 | 0.19 | 0.42 | 0.000 | 0.031 |
| 4 | -59.80 | -61.50 | -60.65 | 1.70 | 0.00 | 0.68 | 0.000 | 0.016 |
| 5 | -46.70 | -58.50 | -52.60 | 11.80 | 0.35 | 0.94 | 0.003 | 0.006 |
| 6 | -56.40 | -60.20 | -58.30 | 3.80 | 0.04 | 0.69 | 0.007 | 0.005 |
| 7 | -44.50 | -58.40 | -51.45 | 13.90 | 0.14 | 0.68 | 0.008 | 0.012 |
| **Mean** | **$-49.34 \pm 9.05$** | **$-55.07 \pm 10.20$** | **$-52.21 \pm 9.28$** | **$5.73 \pm 5.26$** | **$0.18 \pm 0.20$** | **$0.68 \pm 0.22$** | **$0.003 \pm 0.004$** | **$0.026 \pm 0.026$** |




**Table 3.** Isotopic fractionation, extremes of the unreacted fraction parameter ($f_{start}$ and $f_{end}$), and reduction rate ($k_r$) in [ppm/(min · mL KNO$_3$)] for the reduction of N$_2$O from *P. fluorescens*.

| Replica # | $\epsilon_\alpha$ (‰) | $\epsilon_\beta$ (‰) | $\epsilon_{bulk}$(‰) | $\epsilon_{SP}$ (‰) | $f_{start}$ | $f_{end}$ | $k_r$ |
|---|---|---|---|---|---|---|---|
| 1 | 9.70 | 7.10 | 8.30 | 2.60 | 1 | 0 | -0.0027 |
| 2 | 3.10 | 4.10 | 3.40 | -1.00 | 1 | 0 | -0.0060 |
| 3 | 11.00 | 7.70 | 9.30 | 3.30 | 1 | 0 | -0.0022 |
| 4 | 3.70 | 5.80 | 4.60 | -2.10 | 1 | 0 | -0.0027 |
| 5 | 18.20 | 11.70 | 14.80 | 6.50 | 1 | 0 | -0.0014 |
| 6 | 17.10 | 12.30 | 14.50 | 4.80 | 1 | 0 | -0.0019 |
| 7 | 9.30 | 6.50 | 6.50 | 2.80 | 1 | 0 | -0.0009 |
| **Mean** | **10.30 ± 5.86** | **7.89 ± 3.04** | **8.77 ± 4.49** | **2.41 ± 3.04** | **1 ± 0** | **0 ± 0** | **-0.003 ± 1.7e$^{-3}$** |

**Table 4.** Isotopic fractionation, extremes of the unreacted fraction parameter ($f_{start}$ and $f_{end}$), and production rate ($k_p$) in [ppm/(min · mL KNO$_3$)] for the production of N$_2$O from *P. chlororaphis*.

| Replica # | $\epsilon_\alpha$ (‰) | $\epsilon_\beta$ (‰) | $\epsilon_{bulk}$(‰) | $\epsilon_{SP}$ (‰) | $f_{start}$ | $f_{end}$ | $k_p$ |
|---|---|---|---|---|---|---|---|
| 1 | -8.70 | -3.10 | -5.90 | -5.60 | 1 | 0 | 0.237 |
| 2 | -5.00 | -1.10 | -3.05 | -3.90 | 1 | 0 | 0.250 |
| 3 | -6.90 | -3.30 | -5.10 | -3.60 | 1 | 0 | 0.215 |
| 4 | -5.40 | -4.40 | -4.95 | -0.90 | 1 | 0 | 0.182 |
| 5 | -7.60 | -4.50 | -6.05 | -3.10 | 1 | 0 | 0.126 |
| **Mean** | **-6.72 ± 1.54** | **-3.28 ± 1.37** | **-5.01 ± 1.20** | **-3.42 ± 1.69** | **1 ± 0** | **0 ± 0** | **0.201 ± 0.049** |