# Peer review of "Continuous measurements of nitrous oxide isotopomers during incubation experiments"

_Biogeosciences, 2017_

## Referee Comment (RC1) · Anonymous Referee #1 · 13 Jun 2017

The study by Winther et al. presents continuous measurements of nitrous oxide isotopomers at concentration levels close to ambient and intends to determine isotope effects for two different bacterial organisms. Especially the endeavour to bring such measurements towards ambient concentration levels is valuable for the scientific community. In this context, this paper is of interest for the broad audience Biogeosciences attracts. There are some flaws, such as 1. The initial NO3- isotopic composition had to be estimated. 2. There is no nitrate balance provided, which would be helpful with regard to constraining f. 3. Concentration of other products, such as NO have not been accounted for. However, the manuscript provides some interesting calculation approaches, and the experiments involving P. chlororaphis are straightforward. All in

all, I recommend acceptance for publication after addressing the below comments.

See some more detailed comments below. Title ok Highlights Not given Abstract P1, L10: Please specify the sentence "the continuous analysis of . . .". The meaning is unclear without knowing the manuscript. Please change reveal to reveals.

Introduction P2,L19: please change to positions, instead of position The objectives and the added value of another experiment on fractionation factors could be more to the point. Materials and Methods P4, L12 and following: I suggest changing "before" to "upstream of". In addition, the text does not comply with Figure1. From Figure 1, it seems like a Nafion and a Magnesium perchlorate / Ascarite trap was used. The Nafion reduces the water vapor to a certain dewpoint (please specity) and the magnesium perchlorate Mg(ClO4)2! removes the remaining water chemically. The Ascarite (this is not Mg(ClO4)2, but sodium hydroxide coated silica!!) removes the CO2!. This section needs to be corrected. P6,L1: the subsection head number is 2.4.1, but there is no 2.4.2, which does not make sense. I suggest numbering the subsection head to 2.5 P7,L20: please change results to result L7,P19/20: with regard to the reference to the supplementary, I suggest changing "unreacted" to "reacted" in P4,L14 of the supplementary (". . .can therefore be calculated as the sum of the immediate product calculated for all reacted fractions of the substrate"), as the accumulated product is the result of the reacted fractions of the substrate. P8,(8): The numerator terms are quite clear, the denominator term is not required to my understanding. Please clarify. P8,L26: Please change to net production. P8,L30: This assumption is not in agreement with your expectations given on P3,L16-20. There it is assumed that d15Nalpha becomes enriched (I agree with this assumption). In general, all N2O isotopic species should become enriched in a situation in which only reduction occurs, as N2O is the substrate in this case, and a normal isotope effect occurs. However, this is not the case in Figures 6A/B. Please comment. P9,L5: the fractionation factor during reduction is varied between 1 and 2, this means only not-normal isotope effects are allowed for the reduction of N2O. A recent review on isotope effects in the N cycle, Denk et al. 2017

in Soil Biol. Biochem. (The nitrogen cycle: A review of isotope effects and isotope modeling approaches), shows that the literature has reported that N2O reduction is associated with a normal isotope effect for d15Nbulk. Please comment why this limitation was necessary. Results

Discussion P11,L2: The statement that the production rate is 10 times higher for P. Chlororaphis is ambiguous, since a net rate is compared to a "gross" rate (assuming direct conversion of NO3- to N2O). Please add this to your interpretation that reaction rate cannot account for the difference in isotopic fractionation alone. P11,L15-17: This is pertinent information. Thank you. I only suggest to change the numbering from 1) and 2) for the results of the DNA comparisons to i) and ii). I was a little confused with the (1)-(4) numbering.

―――――――――――――――――――――

---

## Referee Comment (RC2) · Anonymous Referee #2 · 30 Jun 2017

The manuscript of Malte Winther et al. describes the real-time analysis of site-specific  $N_2O$  isotopic composition from two denitrifying bacterial strains with a novel Picarro CRDS analyser. A setup for a closed-loop experiment was designed and applied in a number of prototype experiments. A correction function was developed for the spectrometer raw data and a modified Rayleigh model applied to derive fractionation factors.

The manuscript is an important contribution to research on N2O isotopes and therefore of interest for a number of readers of Biogeosciences. The presented interpretation of singular incubation experiments might be questionable; at least given the "surprising" results, e.g. for  $\epsilon_{SP}$  of N2O reduction. But the manuscript should still be accepted after a number of minor revisions as detailed below:

Page 1 Line 8: The main application of the instrument might be for biogeochemical applications, e.g. soil sciences, at enhanced concentrations and not for atmospheric chemistry.

Page 1 Line 10 – 11: The expression "... reveal the transient pattern" is incomplete.

Page 1 Line 15 – 17: The explanation for the SP isotopic fractionation for  $N_2O$  reduction above zero, "diffusive isotopic fractionation and a difference in active enzymes during production of  $N_2O$ ", is not convincing.

Page 2 Line 18 – 19: Please rephrase the sentence "The position in the N2O molecule are named ... " to "N2O molecules with 15N substitution in the central or terminal position are named 15N $\alpha$  for 14N15N16O or 15N $\beta$  for 15N14N16O, respectively."

Page 2 Line 25: Please rephrase the expression to "... to enable continuous and selective measurements of the isotopomer abundances."

Page 3 Line 4 – 6: The sentences "The primary anthropogenic sources of  $N_2O$  are organic and inorganic N fertilizers used for agriculture. The natural sources are primarily nitrification and denitrification in terrestrial and aquatic ecosystems." are misleading as the biotic (and abiotic) source processes for anthropogenic and natural  $N_2O$  emissions are similar, but anthropogenic emissions are enhanced due to fertilizer application. Please rephrase the sentences.

Page 3 Line 17 – 18: The expression "..., the cleavage of N2O is expected to have an increased fractionation effect on 15N $\alpha$ , due to ..." might be rephrased to "the cleavage of N2O is expected to fractionate in favour of the 15N $\alpha$  molecule, due to ...".

Page 3 Line 17 – 19: The statement that "diffusion into the cell (Tilsner et al., 2003) and enzymatic reduction (Wrage et al., 2004)" might be deleted.

Page 4 Line 8: The phrase "by placing the sample delivery system ... in a closed loop" might be rephrased to "by a closed-loop gas flow through the ...".

Page 4 Line 13:  $Mg(CIO_4)_2$  is the chemical formula for Magnesiumperchlorate used for drying the measuring gas, but not for Ascarite used for removing  $CO_2$ . The scheme in Figure 1 shows the correct setup of the trap.

Page 4 Line 22 – 29: Please re-write this section, as the same information, that a concentration dependent correction for delta values is needed is given several times.

Page 5 Line 5: The section on the O2 correction (now Page 6 Line 18 - 23) could better be placed here.

Page 5 Section calibration gases (Table 1): Please check whether there is a mistake in the mean values in Table 1, e.g. the mean of 1.34, 1.08, 2.62 is not 1.32.

Page 6 Line 6: The statement to give  $\delta^{15}N^{\alpha}$  and  $\delta^{15}N^{\beta}$  values for KNO3 is wrong or at least a misunderstanding.

Page 12 Line 3: The statement that differences in net production rates affect  $\epsilon_{SP}$  seems questionable.

Page 12 Line 26: The statement that higher  $\epsilon_{SP}$  values as reported in literature could be rationalized by diffusive isotope fractionation seems questionable, as diffusion is generally assumed to not affect the N2O SP.

Page 13 Line 2: The term "isotope depletion" is incomplete, it should be mentioned which isotopic species is depleted.

---

## Author Response (AR1)

**Response to the referees**

Dear Kees Jan van Groenigen,

In the following document we comment on and explain how we address the issues and comments raised by the two referees. We found the comments of the referees very useful in highlighting important points, missing in the original manuscript. We will take the raised issues into account and adjust the manuscript accordingly. We are grateful and appreciate the two referees for their comments which we believe leads to an improved version of the manuscript.

Thank you for the consideration,

**Response to referee comment # 1**

**General comments:**
The study by Winther et al. presents continuous measurements of nitrous oxide isotopomers at concentration levels close to ambient and intends to determine isotope effects for two different bacterial organisms. Especially the endeavour to bring such measurements towards ambient concentration levels is valuable for the scientific community. In this context, this paper is of interest for the broad audience Biogeosciences attracts. There are some flaws, such as 1. The initial $NO_3^-$ isotopic composition had to be estimated. 2. There is no nitrate balance provided, which would be helpful with regard to constraining f. 3. Concentration of other products, such as NO have not been accounted for. However, the manuscript provides some interesting calculation approaches, and the experiments involving P. chlororaphis are straightforward. All in all, I recommend acceptance for publication after addressing the below comments.

Specific comments:

See some more detailed comments below.
Title ok

Highlights Not given

Abstract P1, L10: Please specify the sentence "the continuous analysis of . . .". The meaning is unclear without knowing the manuscript.
The sentence has been changed to "the continuous measurements of …". (New version P.1, L10-11).

Please change reveal to reveals.
The spelling error has been corrected. (New version P.1, L11).

Introduction P2, L19: please change to positions, instead of position.
The spelling error has been corrected. (New version P.2, L18).

The objectives and the added value of another experiment on fractionation factors could be more to the point.
We added a phrase at the end of the introduction. "Isotope effects during denitrification are diverse and species dependent (e.g. Denk et al., 2017). Our study demonstrates a new way to determine fractionation factors from continuous measurements of $N_2O$." (New version P.3, L23-24).

Materials and Methods P4, L12 and following: I suggest changing "before" to "upstream of".
Agreed and changed. (New version P.4, L11).

In addition, the text does not comply with Figure1. From Figure 1, it seems like a Nafion and a Magnesium perchlorate / Ascarite trap was used. The Nafion reduces the water vapor to a certain dewpoint (please specity) and the magnesium perchlorate $Mg(ClO_4)_2$! removes the remaining water chemically. The Ascarite (this is not $Mg(ClO_4)_2$, but sodium hydroxide coated silica!!) removes the CO2!. This section needs to be corrected.
We agree that the description is incorrect and changed accordingly to: "Before the analyzer, a Nafion unit and an Ascarite trap is installed. The Nafion unit removes the bulk of $H_2O$ vapor. Remaining water vapor is removed chemically by magnesium perchlorate ($Mg(ClO_4)_2$) in front and after the Ascarite (NaOH) section of the trap." (New version P.4, L11-14).

P6, L1: the subsection head number is 2.4.1, but there is no 2.4.2, which does not make sense. I suggest numbering the subsection head to 2.5
Head number has been changed. (New version P.6, L1).

P7, L20: please change results to result
The spelling error has been corrected. (New version P.7, L20).

L7,P19/20: with regard to the reference to the supplementary, I suggest changing "unreacted" to "reacted" in P4,L14 of the supplementary (". . .can therefore be calculated as the sum of the immediate product calculated for all reacted fractions of the substrate"), as the accumulated product is the result of the reacted fractions of the substrate.
We agree that the formulation is confusing and simplify to: "…be calculated as the sum of the respective immediate products." (New supplementary version P.4, L14).

P8, (8): The numerator terms are quite clear, the denominator term is not required to my understanding. Please clarify.
We verified and the mass balance is correct.

P8, L26: Please change to net production.
We agree and corrections has been made accordingly. (New version P.8, L26).

P8, L30: This assumption is not in agreement with your expectations given on P3, L16-20. There it is assumed that $\delta^{15}N^{\alpha}$ becomes enriched (I agree with this assumption). In general, all $N_2O$ isotopic species should become enriched in a situation in which only reduction occurs, as $N_2O$ is the

substrate in this case, and a normal isotope effect occurs. However, this is not the case in Figures 6A/B. Please comment.

We see the origin of the confusion. The outcome of our analysis for *P. fluorescens* is indeed conflicting with the expectation given on P3, L16-20. This is obvious in Figure 6A/B and discussed later in the manuscript. On P8, L30 we only define the start of reduction for determining the fractionation coefficient. The bracket stating "(assumption based on reduction…)" is misleading and we removed it.

P9, L5: the fractionation factor during reduction is varied between 1 and 2, this means only not-normal isotope effects are allowed for the reduction of $N_2O$. A recent review on isotope effects in the N cycle, Denk et al. 2017 in Soil Biol. Biochem. (The nitrogen cycle: A review of isotope effects and isotope modeling approaches), shows that the literature has reported that $N_2O$ reduction is associated with a normal isotope effect for $\delta^{15}N^{bulk}$. Please comment why this limitation was necessary.

We apologize, this is a typo. The reduction fractionation factor was varied between 0 and 2 which includes all possible isotope effects. Corrected. (New version P.9, L5).

Results Discussion P11, L2: The statement that the production rate is 10 times higher for P. Chlororaphis is ambiguous, since a net rate is compared to a "gross" rate (assuming direct conversion of $NO_3^-$ to $N_2O$). Please add this to your interpretation that reaction rate cannot account for the difference in isotopic fractionation alone.

The reviewer is correct. We changed "production" to "net production". (New version P.11, L2).

P11, L15-17: This is pertinent information. Thank you. I only suggest to change the numbering from 1) and 2) for the results of the DNA comparisons to i) and ii). I was a little confused with the (1)-(4) numbering.

We agree and have changed accordingly. (New version P.11, L16).

**Response to referee comment # 2**

The manuscript of Malte Winther et al. describes the real-time analysis of site-specific $N_2O$ isotopic composition from two denitrifying bacterial strains with a novel Picarro CRDS analyser. A setup for a closed-loop experiment was designed and applied in a number of prototype experiments. A correction function was developed for the spectrometer raw data and a modified Rayleigh model applied to derive fractionation factors.

The manuscript is an important contribution to research on $N_2O$ isotopes and therefore of interest for a number of readers of Biogeosciences. The presented interpretation of singular incubation experiments might be questionable; at least given the "surprising" results, e.g. for $\varepsilon_{SP}$ of $N_2O$ reduction. But the manuscript should still be accepted after a number of minor revisions as detailed below:

Page 1 Line 8: The main application of the instrument might be for biogeochemical applications, e.g. soil sciences, at enhanced concentrations and not for atmospheric chemistry.

We agree that the manuscript would be good in soil sciences as well.

Page 1 Line 10 – 11: The expression "… reveal the transient pattern" is incomplete.
Changed to: "The continuous analysis of $N_2O$ isotopomers reveals the transient isotope exchange between $KNO_3$, $N_2O$, and $N_2$." (New version P.1, L10-11).

Page 1 Line 15 – 17: The explanation for the SP isotopic fractionation for $N_2O$ reduction above zero, "diffusive isotopic fractionation and a difference in active enzymes during production of $N_2O$", is not convincing.
We remove the statement (see also our comment to Page 12 Line 26).

Page 2 Line 18 – 19: Please rephrase the sentence "The position in the $N_2O$ molecule are named …" to "$N_2O$ molecules with $^{15}N$ substitution in the central or terminal position are named $^{15}N^{\alpha}$ for $^{14}N^{15}N^{16}O$ or $^{15}N^{\beta}$ for $^{15}N^{14}N^{16}O$, respectively."
The suggested correction is incorrect. Since it is the positions of the N atom which defines the name, and not the molecule. Changed to: "The positions in the $N_2O$ molecule have been named $N^{\alpha}$ and $N^{\beta}$ or short $\alpha$ and $\beta$ (Yoshida and Toyoda, 2000). $N_2O$ molecules with $^{15}N$ substitution in the central or terminal position are named $^{15}N^{\alpha}$ for $^{14}N^{15}N^{16}O$ or $^{15}N^{\beta}$ for $^{15}N^{14}N^{16}O$, respectively." (New version P.2, L16-19).

Page 2 Line 25: Please rephrase the expression to "… to enable continuous and selective measurements of the isotopomer abundances."
Correction has been applied. (New version P.2, L24).

Page 3 Line 4 – 6: The sentences "The primary anthropogenic sources of $N_2O$ are organic and inorganic N fertilizers used for agriculture. The natural sources are primarily nitrification and denitrification in terrestrial and aquatic ecosystems." are misleading as the biotic (and abiotic) source processes for anthropogenic and natural $N_2O$ emissions are similar, but anthropogenic emissions are enhanced due to fertilizer application. Please rephrase the sentences.
The sentences has been corrected. Now it reads "The primary anthropogenic increase in $N_2O$ emission originate from organic and inorganic N fertilizers used for agriculture. The natural sources are primarily nitrification and denitrification in terrestrial and aquatic ecosystems." (New version P.3, L1-3).

Page 3 Line 17 – 18: The expression "… , the cleavage of $N_2O$ is expected to have an increased fractionation effect on $^{15}N^{\alpha}$, due to …" might be rephrased to "the cleavage of $N_2O$ is expected to fractionate in favor of the $^{15}N^{\alpha}$ molecule, due to …".
We adapt the suggestion and changed accordingly. (New version P.3, L14-15).

Page 3 Line 17 – 19: The statement that "diffusion into the cell (Tilsner et al., 2003) and enzymatic reduction (Wrage et al., 2004)" might be deleted.
These introductory statements line out what ideas have been presented to explain changes in SP. We would like to keep them.

Page 4 Line 8: The phrase "by placing the sample delivery system … in a closed loop" might be rephrased to "by a closed-loop gas flow through the …".
We adapt the suggestion and changed accordingly. (New version P.4, L8).

Page 4 Line 13: $Mg(ClO_4)_2$ is the chemical formula for Magnesiumperchlorate used for drying the measuring gas, but not for Ascarite used for removing $CO_2$. The scheme in Figure 1 shows the

correct setup of the trap.

We agree and the correction has been made accordingly. Also see comments to referee #1. (New version P.4, L11-14).

Page 4 Line 22 – 29: Please re-write this section, as the same information, that a concentration dependent correction for delta values is needed is given several times.

We removed duplicate statements from the section and write now: "Isotopomer measurements made with the G5101i-CIC have a $N_2O$ concentration dependence and need to be corrected. There is a 1/concentration dependence, caused by small offsets in the measurement of the $^{14}N^{15}N^{16}O$ and $^{15}N^{14}N^{16}O$ peaks. These offsets are caused by baseline ripple created by optical cavity etalons. An etalon is an optical effect in which a beam of light undergoes multiple reflections between two reflecting surfaces, and whose resulting optical transmission or reflection is periodic in wavelength. The ripples are not always constant in phase, which means that the ripples can shift spectrally, which can cause the offset to drift over time. Because baseline ripple effects become more dominant as $N_2O$ concentration decreases, the offset is largest at low concentrations." (New version P.4, L22-28).

Page 5 Line 5: The section on the $O_2$ correction (now Page 6 Line 18 – 23) could better be placed here.

The $O_2$ correction is related to the calibration gases we use for the specific experiments introduced on page 6. Therefore we prefer to leave it where it is. Moving the section to page 5 would lead to repeating the arguments on page 6.

Page 5 Section calibration gases (Table 1): Please check whether there is a mistake in the mean values in Table 1, e.g. the mean of 1.34, 1.08, 2.62 is not 1.32.

The mean values in Table 1 is not the mean of the three numbers, but rather the combined mean values, which depends on the number of measurements performed. That is the reason for the difference. That we use the combined mean values has been clarified.

Page 7 Line 6: The statement to give $\delta^{15}N^{\alpha}$ and $\delta^{15}N^{\beta}$ values for $KNO_3$ is wrong or at least a misunderstanding.

We agree that the way we phrased is misleading and reformulated to clarify the link between $\delta^{15}N^{\alpha}$, $\delta^{15}N^{\beta}$ and the isotopic composition of $KNO_3$: "The initial isotopic composition of $KNO_3$ calculates as the average of the end values for $\delta^{15}N^{\alpha}$ and $\delta^{15}N^{\beta}$ to -3.08 ‰ ± 1.05 (identical to the $\delta^{15}N^{bulk}$ value)." (New version P.7, L5-7).

Page 12 Line 3: The statement that differences in net production rates affect $\varepsilon_{SP}$ seems questionable.
We agree and also write earlier in the manuscript that net production differences account for less than 10% of the effect. We now write: "We hypothesize that the slight difference in $\varepsilon_{SP}$ originates predominantly from 4) fractionation associated with nitrous oxide reductase in *P. fluorescens*." (New version P.12, L5-6).

Page 12 Line 26: The statement that higher $\varepsilon_{SP}$ values as reported in literature could be rationalized by diffusive isotope fractionation seems questionable, as diffusion is generally assumed to not affect the $N_2O$ SP.

We agree that diffusive isotope fractionation is mass dependent and therefore has no effect on SP. Rereading the section we realize that the reduction part of Figure 7 is not fully described and would

therefore like to slightly adjust the phrasing. As to the statement in question it is misplaced and we remove it. The paragraph now reads: "A number of studies have investigated $N_2O$ reduction from denitrification in soils (e.g. (Well and Flessa, 2009b; Köster et al., 2013a; Lewicka-Szczebak et al., 2014, 2015)). The results are only partly in accord with our findings for specific bacteria strains. While our results for $\varepsilon_{bulk}$ are within the range of their findings, they find consistently negative $\varepsilon_{SP}$ values while our results are generally positive. The only study on pure bacteria we know of is from Ostrom et al. (2007) for two bacteria strains different from ours namely *P. stutzeri* and *P. denitrificans*. They found $\varepsilon_{SP}$ values between -6.8 ‰ and -5 ‰. At this point we have no explanation for the discrepancy but can find no artifact in our incubation setup." (New version P.12, L23-28).

Page 13 Line 2: The term "isotope depletion" is incomplete, it should be mentioned which isotopic species is depleted.

We agree and have adapted the formulation. Now we write "… find a bulk isotope depletion." (New version P.13, L4).

[revised manuscript text omitted]

Measurements are made by a closed-loop gas flow through the G5101i-CIC with a microbial incubation glass chamber (Fig. 1). Circulation is provided by a leak-reduced diaphragm pump installed downstream from the analyzer. The pump (KNF N84.4 ANE) has been sealed using vacuum sealant (Celvaseal high vacuum leak sealant, Myers vacuum repair service, Inc., Kittanning, PA 16201, USA). A low leak rate is prerequisite for accurate measurements in a closed loop experiment. Upstream of the analyzer, a Nafion unit and an Ascarite trap is installed. The Nafion unit removes the bulk of $H_2O$ vapor. Remaining water vapor is removed chemically by magnesium perchlorate($Mg(ClO_4)_2$) in front and after the Ascarite (NaOH) section of the trap. 
[revised manuscript text omitted]

25  results are generally positive. The only study on pure bacteria we know of is from Ostrom et al. (2007) for two bacteria strains different from ours namely *P. stutzeri* and *P. denitrificans*. They found $\epsilon_{SP}$ values between -6.8 ‰ and -5 ‰. At this point we have no explanation for the discrepancy but can find no artifact in our incubation setup.

**5 Conclusions**

30  We have presented successful continuous measurements of the denitrifying bacterial process using two different strains of bacteria: *P. fluorescens* which is a full denitrifier, and *P. chlororaphis* which is a denitrifier without nitrous oxide reductase activity. Assuming a Rayleigh type fractionation, modified for isotopomers and simultaneous reduction, we have calculated the isotopic

fractionation during production and reduction of $N_2O$. The isotopic fractionation for *P. chlororaphis* is in line with previous results for both SP and bulk. For *P. fluorescens*, we find similar SP isotopic fractionation values during $N_2O$ production, while we find slightly increased SP isotopic fractionation values during $N_2O$ reduction. The bulk isotopic fractionation calculated for $N_2O$ reduction is in line with previously presented results though for production we find a bulk isotope depletion. We believe

5    that, in our experiment, the bulk isotope depletion is due to a difference in production rates, in nitrite reductase, and in nitrous oxide reductase.

*Author contributions.* MW and TB designed the experiments and MW carried out the measurements and analyzed data. SC prepared the bacteria prior to experiments. AP analyzed the bacterial DNA sequences. DBH and EC developed the G5101i-CIC analyzer. MW prepared the manuscript with contribution from all co-authors.

10   *Acknowledgements.* We thank Jan Kaiser for isotope specific gas samples used for our reference gases, Sakae Toyoda, Naohiro Yoshida, Carina van der Veen, and Thomas Röckmann for assistance on measurements of our reference gases. We want to thank Center for Permafrost (CENPERM DNRF100) and the Centre for Ice and Climate, funded by the Danish National Research Foundation for their support, and The Danish Agency for Science Technology and Innovation, for funding used in supporting this project. We want to thank Picarro Inc. and especially Eric Crosson, and Nabil Saad for the collaboration and guidance in the development of the $N_2O$ isotope analyzer prototype used

15   in this project.

The presented bacterial experiments are performed under anaerobic conditions, where oxygen ($O_2$) is not present. The effect of this lack of $O_2$ on the isotopic signal was quantified by performing dilution experiments on the two CIC-MPI standard gasses. Two stepwise dilution measurement with either pure synthetic air or $N_2$ was conducted for each of the standard gases. The $N_2$ was of purity 99.9999 % and the synthetic air was a $N_2/O_2$ mixture (20.1 % $O_2$ and 79.9 % $N_2$, purity 99.999 %). Figure S1 show the average measured $\delta^{15}N^\alpha$ values for each step during each of the four four dilution experiments. Similar measurements were performed on $\delta^{15}N^\beta$ and $\delta^{15}N^{bulk}$. The two dilution experiments with synthetic air (Fig. S1A and S1B) results in a dependence only on the $N_2O$ concentration. The two dilution experiments with $N_2$ (Fig. S1C and S1D) results in a dependence on both the $N_2O$ concentration and the $O_2$ concentration.

[Figure]

**Figure S1.** *The results of the four dilution experiments for $\delta^{15}N^\alpha$. (A) CIC-MPI-II diluted with synthetic air (B) CIC-MPI-I diluted with synthetic air, (C) CIC-MPI-II diluted with $N_2$, and (D) CIC-MPI-I diluted with $N_2$. Black points represent the mean of 30 minutes continuous measurements. The standard error (error bars) of the measurements are shown in red.*

The effect of a changing $O_2$ concentration on the isotopic composition of $N_2O$ is assessed from the difference between the two dilution experiments for each standard gas. When calculating the difference between the two dilution experiments we isolate the dependence on the $O_2$ concentration, i.e. the difference in $\delta^{15}N^\alpha$ is plotted versus the $O_2$ concentration. Figure S2 show the difference between dilution experiments performed with synthetic air and $N_2$. I.e. the difference between Fig. S1A and Fig. S1C for CIC-MPI-II and the difference between Fig. S1B and Fig. S1D for CIC-MPI-I. Similar measurements were performed on $\delta^{15}N^\beta$ and $\delta^{15}N^{bulk}$.

[Figure]

**Figure S2.** *The results of the difference between dilution experiments performed with synthetic air and $N_2$ performed on $\delta^{15}N^\alpha$. (A) difference measured on CIC-MPI-I, and (B) difference measured on CIC-MPI-II. Black points represent the mean of 30 minutes continuous measurements. The standard error (error bars) of the measurements are shown in red for both the $\delta$-value and the $O_2$ concentration.*

The average response of the isotope composition with respect to the $O_2$ concentration averaged for the two standard gasses are presented in Fig. S3. This relation is calculated using a Monte Carlo algorithm applied to a linear relation model for the $O_2$ concentration and $\delta^{15}N^\alpha$, $\delta^{15}N^\beta$, and $\delta^{15}N$ respectively.

The linear relation model for the effect of $O_2$ concentration ($[O_2]$) to the values of $\delta^{15}N^\alpha$, $\delta^{15}N^\beta$, and $\delta^{15}N$ are shown in equation 1, 2, and 3, respectively.

$$\delta^{15}N^\alpha_{ODC} = -0.1534 \cdot [O_2] + 15.27 \tag{1}$$

[Figure]

**Figure S3.** *The effect of changing $O_2$ concentration on the (A) $\delta^{15}N^{\alpha}$, (B) $\delta^{15}N^{\beta}$, and (C) $\delta^{15}N$. Black points represent the response of the isotope composition with respect to the $O_2$ concentration averaged for the two standard gasses. The standard error (error bars) of the measurements are shown in black for both the $\delta$-value and the $O_2$ concentration. The blue lines are mean values of the Monte Carlo simulation. The green lines are the 1-$\sigma$ error-bar calculated using Monte Carlo simulation.*

$$\delta^{15}N^{\beta}_{ODC} = -0.1210 \cdot [O_2] + 11.66 \tag{2}$$

$$\delta^{15}N^{bulk}_{ODC} = -0.1372 \cdot [O_2] + 13.47 \tag{3}$$

5  In the above $\delta^{15}N^{\alpha}_{ODC}$, $\delta^{15}N^{\beta}_{ODC}$, and $\delta^{15}N^{bulk}_{ODC}$ represent the offsets which are subtracted from the $N_2O$ concentration corrected data.

The bacterial evolution experiments were performed under pure $N_2$ conditions. The reported values in the article, are derived by first applying the $N_2O$ concentration correction and then subsequently applying the correction for the lack of $O_2$. With zero $O_2$ this correction becomes a simple subtraction by the intercept values of the above linear models.

**1.2 Rayleigh model for isotopomers of $N_2O$**

Mariotti et al. (1981) derived the equation for the isotope ratio of the substrate as:

$$\frac{R_s}{R_{s,0}} = f^{(\alpha_{p/s}-1)} \tag{4}$$

where $R_{s,0}$ is the initial isotope ratio of the substrate, $R_s$ is the isotope ratio of the substrate at time $t$, $\alpha_{p/s}$ is the fractionation factor of the product versus the substrate, and $f$ is the unreacted fraction of substrate at time $t$. I.e. $f$ is going in steps from 1 to 0 during the reaction. The fractionation factor of the product versus the substrate is the bulk fractionation factor $\alpha_{bulk}$, and therefore $\alpha_{bulk} = (\alpha_\alpha + \alpha_\beta)/2$.

The fractionation factor is a constant calculated as $\alpha_{p/s} = R_p/R_s$. The immediate product for the two isotopomers ($R_{im}^i$) therefore calculates as:

$$R_{im}^i = R_s \cdot \alpha_i \tag{5}$$

$$\Updownarrow$$

$$R_{im}^i = \alpha_i \cdot R_{s,0} \cdot f^{(\alpha_{bulk}-1)} \tag{6}$$

where the isotopomers are distinguished with $i$ ($i = 1, 2$), respectively. The accumulated product for each of the isotopomers ($R_{p,acc}^i$) can therefore be calculated as the sum of the respective immediate products.

$$R_{p,acc}^i = \frac{1}{1-f} \int_f^1 R_{im}^i df' \tag{7}$$

$$\Updownarrow$$

$$R_{p,acc}^i = \frac{1}{1-f} \int_f^1 \alpha_i \cdot R_{s,0} \cdot f'^{(\alpha_{bulk}-1)} df'. \tag{8}$$

where $f$ is the unreacted fraction of the substrate. The equation for the accumulated product for each of the isotopomers derives to:

$$R_{p,acc}^i = \alpha_i \cdot \frac{R_{s,0}}{1-f} \int_f^1 f'^{(\alpha_{bulk}-1)} df \tag{9}$$

$$\Downarrow$$

$$R_{p,acc}^i = \alpha_i \cdot \frac{R_{s,0}}{1-f} \cdot \left[ \frac{f'^{\alpha_{bulk}}}{\alpha_{bulk}} \right]_f^1 \tag{10}$$

$$\Downarrow$$

$$R_{p,acc}^i = \frac{\alpha_i}{\alpha_{bulk}} \cdot R_{s,0} \cdot \frac{1-f^{\alpha_{bulk}}}{1-f} \tag{11}$$

$$\Downarrow$$

$$R_{p,acc}^i = \frac{\alpha_i}{\alpha_{bulk}} \cdot R_{p,acc}^{bulk}. \tag{12}$$

The isotopomer correction factors for the two isotopomers ($\varphi_\alpha$ and $\varphi_\beta$) therefore ends up as presented in the manuscript.

$$\varphi_\alpha = \frac{\alpha_\alpha}{\alpha_{bulk}} \quad , \quad \varphi_\beta = \frac{\alpha_\beta}{\alpha_{bulk}} \tag{13}$$

**1.3 Iterative determination of unreacted fraction ($f$)**

**1.3.1 Figures of $f_{start}$ for $\delta^{15}N^{\alpha}$**

Figures of the iterative determination of the starting point of the unreacted fraction ($f$) of $\delta^{15}N^{\alpha}$ versus the calculated R-squared value. The black crosses are all possible $f$-values used in calculation of $R^2$ between the Rayleigh fractionation profile and the measured data produced from *Pseudomonas fluorescens*. The red crosses are the best fit to $\delta^{15}N^{\alpha}$. The green crosses are the average best fit to $\delta^{15}N^{\alpha}$ and $\delta^{15}N^{\beta}$, hence the used values. Figure A is the first replica and the one presented in the manuscript.

[Figure]

**1.3.2 Figures of $f_{start}$ for $\delta^{15}N^{\beta}$**

Figures of the iterative determination of the starting point of the unreacted fraction ($f$) of $\delta^{15}N^{\beta}$ versus the calculated R-squared value. The black crosses are all possible $f$-values used in calculation of $R^2$ between the Rayleigh fractionation profile and the measured data produced from *Pseudomonas fluorescens*. The red crosses are the best fit to $\delta^{15}N^{\beta}$. The green crosses are the average best fit to $\delta^{15}N^{\alpha}$ and $\delta^{15}N^{\beta}$, hence the used values. Figure A is the first replica and the one presented in the manuscript.

[Figure]

**1.3.3   Figures of $f_{end}$**

Figures of the iterative determination of the ending point of the unreacted fraction ($f$) versus the calculated R-squared value. The black crosses are all possible $f$-values used in calculation of $R^2$ between the Rayleigh fractionation profile and the measured data produced from *Pseudomonas fluorescens*. The green crosses are the best fit to $\delta^{15}N^{\alpha}$ and $\delta^{15}N^{\beta}$, hence the used values. Figure A is the first replica and the one presented in the manuscript.

[Figure]

**1.4 Iterative determination of the reduction correction parameter ($\gamma$)**

Figures of the iterative determination of the starting point of the reduction correction parameter ($\gamma$) versus the calculated R-squared value. The black crosses are all possible $\gamma$-values used in calculation of $R^2$ between the Rayleigh fractionation profile and the measured data produced from *Pseudomonas fluorescens*. The green crosses are the best fit to $\delta^{15}N^{\alpha}$ and $\delta^{15}N^{\beta}$, hence the used values. Figure A is the first replica and the one presented in the manuscript.

[Figure]

**1.5 Pseudomonas Chlororaphis**

**1.5.1 Figures of $\delta^{15}N^\alpha$**

Figures of the continuous measurements of the evolution of $\delta^{15}N^\alpha$ versus the concentration of $N_2O$. The blue profile is the raw production part. The black profile is the five minutes running mean of the raw measurements. The red is the fitted Rayleigh distillation for the production part. Figure A is the first replica and the one presented in the manuscript.

[Figure]

**1.5.2 Figures of $\delta^{15}N^\beta$**

Figures of the continuous measurements of the evolution of $\delta^{15}N^\beta$ versus the concentration of $N_2O$. The blue profile is the raw production part. The black profile is the five minutes running mean of the raw measurements. The red is the fitted Rayleigh distillation for the production part. Figure A is the first replica and the one presented in the manuscript.

[Figure]

**1.5.3 Figures of $\delta^{15}N^{bulk}$**

Figures of the continuous measurements of the evolution of $\delta^{15}N^{bulk}$ versus the concentration of $N_2O$. The blue profile is the raw production part. The black profile is the five minutes running mean of the raw measurements. The red is the fitted Rayleigh distillation for the production part. Figure A is the first replica and the one presented in the manuscript.

[Figure]

**1.5.4 Figures of SP**

Figures of the continuous measurements of the evolution of SP versus the concentration of $N_2O$. The blue profile is the raw production part. The black profile is the five minutes running mean of the raw measurements. Figure A is the first replica and the one presented in the manuscript.

[Figure]

**1.6 Pseudomonas Fluorescens**

**1.6.1 Figures of $\delta^{15}N^{\alpha}$**

Figures of the continuous measurements of the evolution of $\delta^{15}N^{\alpha}$ versus the concentration of $N_2O$. The blue profile is the raw production part. The green profile is the raw consumption part. The black profile is the 5 minutes running mean of the raw measurements. The red is the fitted Rayleigh distillation for the production part. The magenta is the fitted Rayleigh distillation for the consumption part. Figure A is the first replica and the one presented in the manuscript.

[Figure]

**1.6.2 Figures of $\delta^{15}N^\beta$**

Figures of the continuous measurements of the evolution of $\delta^{15}N^\beta$ versus the concentration of $N_2O$. The blue profile is the raw production part. The green profile is the raw consumption part. The black profile is the five minutes running mean of the raw measurements. The red is the fitted Rayleigh distillation for the production part. The magenta is the fitted Rayleigh distillation for the consumption part. Figure A is the first replica and the one presented in the manuscript.

[Figure]

**1.6.3 Figures of $\delta^{15}N^{bulk}$**

Figures of the continuous measurements of the evolution of $\delta^{15}N^{bulk}$ versus the concentration of N$_2$O. The blue profile is the raw production part. The green profile is the raw consumption part. The black profile is the five minutes running mean of the raw measurements. The red is the fitted Rayleigh distillation for the production part. The magenta is the fitted Rayleigh distillation for the consumption part. Figure A is the first replica and the one presented in the manuscript.

[Figure]

**1.6.4 Figures of SP**

Figures of the continuous measurements of the evolution of SP versus the concentration of $N_2O$. The blue profile is the raw production part. The green profile is the raw consumption part. The black profile is the five minutes running mean of the raw measurements. Figure A is the first replica and the one presented in the manuscript.